# Sub-micro porous thin polymer membranes for discriminating $H_2$ and $CO_2$

Xueru Yan[1,2], Tianqi Song[3], Min Li[1,2], Zhi Wang [1,2] & Xinlei Liu [1,2] ✉

Polymeric membranes with high permeance and remarkable selectivity for simultaneous $H_2$ purification and $CO_2$ capture under industry-relevant conditions are absent. Herein, sub-micro pores with precise molecular sieving capability are created in ultra-thin (13–30 nm) polymer membranes via controllable transformation of amine-linked polymer (ALP) films into benzimidazole-and-amine-linked polymer (BIALP) layers. The BIALP membranes exhibit stable unprecedented $H_2/CO_2$ selectivity of 120 with a $H_2$ permeance of 315 GPU. Furthermore, high pressure (up to 11 bar) and thermal (up to 300 °C) resistance is delivered. This work provides a concept on designing porous polymeric membranes for precise molecular discrimination.

Membrane technology is promising for precise separation of small molecules, which will have a significant impact on industrial production[1,2]. Such challenging separation requires membranes with superior capability for discrimination, for instance, precise molecular sieving[3,4]. Polymer membranes with suitable pores could tackle this challenge[5,6]. The design and synthesis of porous polymer membrane materials have been investigated recently, including thermally rearranged (TR) polymers[7,8], polymers of intrinsic microporosity (PIMs)[9–11], and porous organic frameworks (POFs)[12–15], using approaches such as stacking 2D porous layers[12], integrating aligned synthesis[16,17], introducing nanocavities[18,19], and post-treatment[20] to design channels. However, for separating small gas pairs, such as $H_2/CO_2$ (kinetic diameter, 0.29/0.33 nm), the membrane performance and preparation technology should be further improved to meet industrial requirements[21].

POFs are porous, organic, network polymers linked by covalent bonds[22]. Benzimidazole-linked polymers (BILPs) are a family of POFs, the linkage of which is benzimidazole[22]. BILP membranes have narrow pores, rendering them the molecular sieving ability for separating small molecules[22–25]. Unfortunately, for $H_2/CO_2$ separation, BILPs and modified BILPs membranes showed moderate gas permeance and selectivity due to their intrinsic rigidity, over-dense packing, and insufficient porosity for $H_2$ transport[26]. Therefore, creating adequate $H_2$-selective pores is desirable. A combination of rigid and flexible polymer segments could provide proper pores for small gas separation[7,27]. A synergy between intrinsic pores offered by rigid chains and transient pores suggested by flexible chains elicits excellent molecular sieving behavior. However, reported typical polymers, like PIMs and TR polymers, exhibit low selectivity for $H_2/CO_2$ separation because of their unsuitable pore size[28–30].

Herein, sub-microporous ultra-thin (down to 13 nm) polymer membranes were fabricated by translating amine-linked polymer (ALP) films into benzimidazole-and-amine-linked polymer (BIALP) layers (Fig. 1a). BIALPs are distinct from BILPs as amine linkages are present in the former ones. Interfacial polymerization (IP) protocol was employed followed by thermal treatment. Relatively flexible amine-linked segments together with rigid benzimidazole-linked parts could effectively generate narrow intrinsic and transient pores (Fig. 1b). Thus, sufficient $H_2$-selective channels were created in ultra-thin BIALP membranes relying on controllable transformation of ALP segments to BILP networks. Therefore, $H_2$ permeance and $H_2/CO_2$ selectivity were boosted simultaneously together with excellent steam stability under elevated temperature and pressure.

## Results

### Fabrication of membranes

IP was selected to in-situ synthesize the membranes, owing to its good reproducibility and facile operation[15,31,32]. In detail, ALP films (Supplementary Movie 1) and membranes were formed by reacting 1,2,4,5-benzenetetramine (BTA) with trimesoyl chloride (TMC) at the interface between aqueous and n-hexane phases (Fig. 1a and Supplementary

[1]Chemical Engineering Research Center, School of Chemical Engineering and Technology, Tianjin University, 300072 Tianjin, China. [2]Tianjin Key Laboratory of Membrane Science and Desalination Technology, Haihe Laboratory of Sustainable Chemical Transformations, State Key Laboratory of Chemical Engineering, Tianjin University, 300072 Tianjin, China. [3]School of Computer Science and Technology, Xi'an Jiaotong University, 710049 Xi'an, China. ✉e-mail: xinlei_liu1@tju.edu.cn

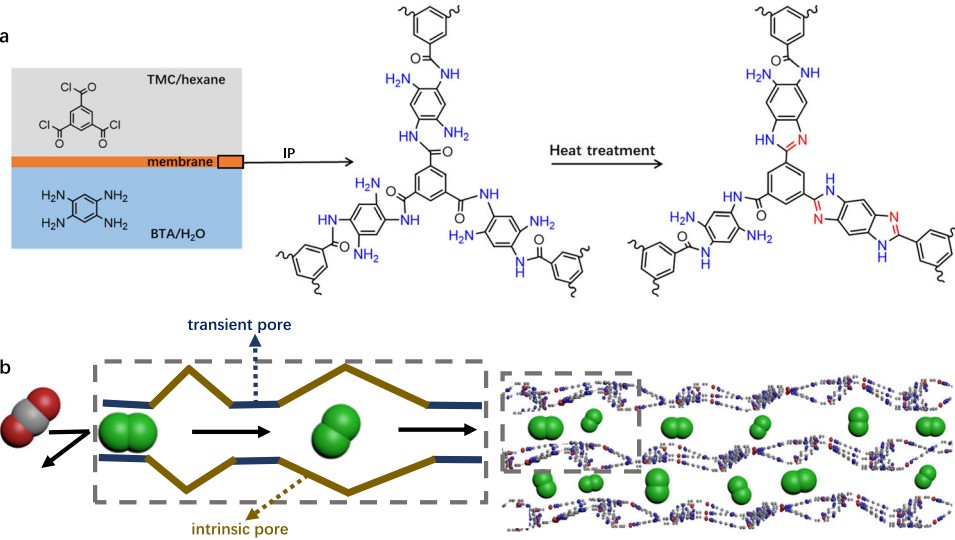

**Fig. 1 | Schematic representation of fabrication method and structure of BIALP membranes. a** IP and thermal treatment for fabricating BIALP membranes. **b** $H_2$-selective channels for $H_2/CO_2$ separation. The left dash rectangle is the enlarged schematic view of the right one.

Fig. 1). The films synthesized on the surface of substrates were named membranes, and photos of the films and BIALP membranes were shown in Supplementary Fig. 1. After thermal treatment, ALPs were partly translated into BILP networks resulting in new polymer chain packing and network structures with enhanced intrinsic sub-micropores (Fig. 1b). Furthermore, the combination of rigid BILP and flexible ALP segments in BIALP membranes efficiently created sub-micro $H_2$-selective transient pores and prohibit over-dense packing of polymer chains in the membranes. The BIALP membranes showed ultra-thin thickness thanks to the combined action of the fast reaction rate of monomers, the small pore size of substrates, and post-treatment at high temperatures. To study the structural properties of these polymers, films, and powders were also prepared for characterization. In detail, preparation procedures were described in the Methods part. The numbers in the names of the polymers synthesized in this work are the temperatures (°C) of thermal treatment and the pH values are measured based on the aqueous phases for IP. The pH value is 1 if not specified.

## Structural properties of membranes

The chemical structure transformation from ALP to BIALP was validated using X-ray photoelectron spectroscopy (XPS). After thermal treatment, a new peak assigned to -N= (398.1 eV) from benzimidazole rings appeared (Fig. 2a, b). More benzimidazole segments were generated at higher temperatures (Supplementary Fig. 2 and Supplementary Table 1). $N1s$ spectra showed a right shift and reduction of -N$^+$ content at higher temperatures, which could be rationalized by the deprotonation of N atoms because of volatilization H$^+$ (in terms of HCl)[33]. The formation of amide and benzimidazole linkages was further corroborated by the feature peaks in $^{13}C$ solid-state nuclear magnetic resonance ($^{13}C$ ss-NMR) spectra (Fig. 2c and Supplementary Fig. 3) at 175 ppm (number 3) and 150 ppm (number 7)[22,23,34], respectively, in line with the XPS data.

The X-ray diffraction (XRD) patterns of ALP and BIALP200 (pH = 1) powders (under ambient condition, Fig. 2d) reflected two types of diffractions, in which, the peaks centered at 11° corresponding to d-spacing of 8.0 Å were due to stacking of the film layers[35]. Whereas the broad diffractions at 24° with d-spacing of 3.7 Å could be assigned to the packing of polymer chains[23]. Owning to the thermal motions of polymer chains, the broad diffractions moved to 27° and provided a d-spacing of 3.3 Å at 150 °C in a mixed $H_2/CO_2$ surrounding (Fig. 2e). The packed polymer chains will generate transient pores for $H_2/CO_2$

separation. The stacking of the film layers became loose, and peaks at 6.7° with d-spacing of 13.3 Å were recognized.

Both ALP and BIALP films could be handled but tore easily. A video of an ALP film pierced by a pipette was attached to the supplementary materials (Supplementary Movie 1). The thickness of BIALP (pH = 1) membranes and films decreased from 30 to 13 nm with temperature of thermal treatment varying from 150 to 300 °C (scanning electron microscope (SEM) images in Supplementary Fig. 4, and atomic force microscope (AFM) images in Fig. 2f, g and Supplementary Fig. 5). More unreacted monomers and oligomers volatilized at higher temperatures (Supplementary Fig. 6), interpreting the phenomenon of thickness change. In addition, temperature could regulate the ratio of amide and benzimidazole linkages, rearrange the packing style of polymer chains, and slightly change the density of membranes.

The calculated Brunauer-Emmett-Teller (BET) area of ALP (11.4 m$^2$ g$^{-1}$) and BIALP200 (pH = 1) (21.5 m$^2$ g$^{-1}$) powders were very low based on $N_2$ adsorption (Fig. 2i), comprising just external surface area according to the t-plot method. This result suggests that $N_2$ molecules had no access to the narrow micropores in these polymers. From Fig. 2h, $CO_2$ adsorption showed a BET area of 83.7 m$^2$ g$^{-1}$ for BIALP200 (pH = 1), almost four times of ALP (21.0 m$^2$ g$^{-1}$). The BIALP200 (pH = 1) showed a relatively higher microporous surface area of 58.9 m$^2$ g$^{-1}$ with a density of 1.56 ± 0.005 g cm$^{-3}$, highlighting the existence of sub-micropores in its internal structure. These micropores will serve as channels for $H_2$ transport at elevated temperatures since the uptake of $CO_2$ drops sharply with temperature[36].

## Structural regulation of membranes via varying pH values

Polymerization reaction between amine and trimesoyl chloride monomers will generate H$^+$ ions as the byproduct[32,37], which will decrease the pH of reserved BTA aqueous solutions. Additionally, amine monomers (BTA) used in this work contain hydrochloric acid for stability. Therefore, the formation of ALP frameworks could be regulated by preadjusting pH to control the reaction rate of IP[38] and trans-interface diffusion of amine monomers[32,37].

The morphology of BIALP membranes was regulated by pre-adjusting the pH of amine solutions (Fig. 3). When the aqueous phase kept high acidity (pH = 1), the IP reaction rate would be retarded because of the H$^+$ byproduct, which provided a peaceful reaction area resulting a smooth surface of BIALP (pH = 1) membrane (Fig. 3a and Supplementary Fig. 7a). According to TEM images, in this case, the membrane layer was well attached to the substrate after slicing with

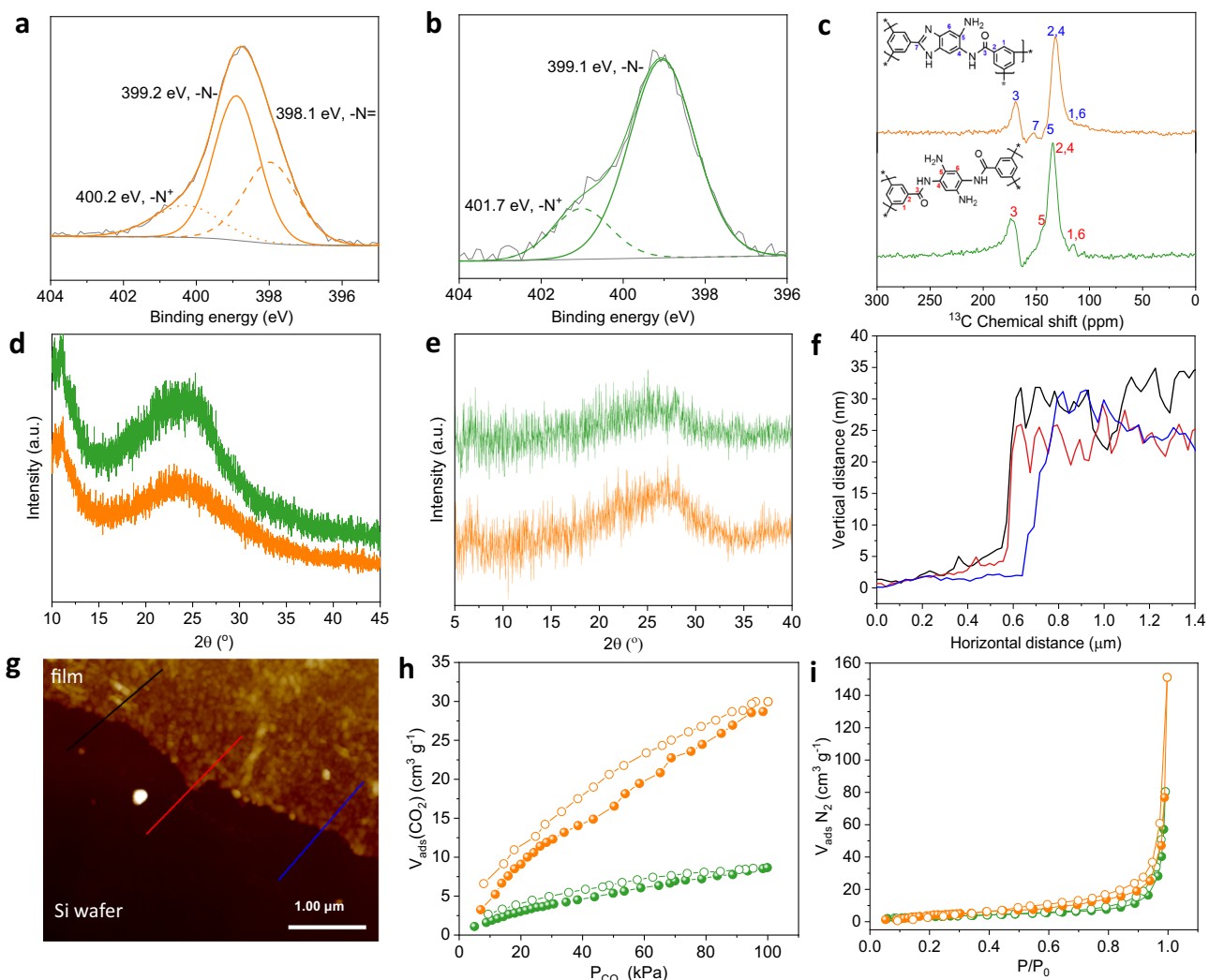

**Fig. 2 | Characterization of membranes, films, and powders. a** Narrow scans of *NIs* spectra of BIALP membrane, and **b** ALP membrane. **c** $^{13}C$ ss-NMR spectra of powders. **d** XRD patterns of powders under ambient conditions, and **e** XRD patterns of powders in a 1 bar $H_2/CO_2$ (1/1, mol/mol) surrounding at 150 °C. **f** Height profile and **g** AFM image of BIALP200 film on the silicon wafer. The three lines in **f** are corresponding by color to the lines in **g**. **h** $CO_2$ adsorption-desorption isotherms of powders at 25 °C. **i** $N_2$ adsorption-desorption isotherms of powders at −196 °C. To guide the eyes, in Fig. 2, green symbols are for ALP materials and orange ones for BIALP200 (pH = 1).

focused ion beam (Fig. 3b, c). When the BTA solution was preadjusted to pH = 8, the BIALP200 (pH = 8) membrane showed a similarly smooth surface to the BIALP200 (pH = 1) membrane (Fig. 3d and Supplementary Fig. 7b). However, voids were generated between the membrane layer and substrate after slicing (Fig. 3e, f), pointing a weaker adhesion. This is because the alkali quickly neutralized the H+ produced, accelerating the reaction and causing BTA to spread deeper in the TMC solution. The roughness of the membrane surface was increased from 5.30 to 9.65 nm (Supplementary Fig. 8a, b). When the BTA solution kept a high alkalinity (pH = 13) that could have a strong buffering capacity for H+ byproducts[33], BTA rapidly reacted with TMC, causing a sharp decrease in local TMC concentrations, releasing heat, and resulting in an unstable reaction zone[33,35]. Therefore, BIALP200 (pH = 13) membranes provided a crumpled surface, relatively high roughness of 20.7 nm (Supplementary Fig. 8c), and larger gaps between the membrane layer and substrate after slicing (Fig. 3g–i and Supplementary Fig. 7c).

A molecular dynamics (MD) system was used to simulate the diffusion of monomers across the water/hexane interface with pH value variation of aqueous solutions. The final concentration of monomers in the reaction area was shown in Fig. 4. The results

demonstrated that when pH values of the BTA solution were 1, fewer BTA molecules crossed the water/hexane interface (pink dashed line) corresponding to lower relative concentration (Fig. 4a). Final snapshots (80 ps) showed that the concentration of BTA and TMC near the water/n-hexane interfaces was relative equilibrium (Fig. 4d). When the alkalinity of BTA solution increased (pH from 8 to 13, Fig. 4b, c), a large number of BTA molecules crossed the interface and reached deeper in TMC solution. A sharp decrease in local TMC concentrations and an uneven distribution of monomers in the reaction region were found (Fig. 4e, f). The surface morphology and microstructure of membranes greatly affect the separation performance (Supplementary Fig. 9). BIALP200 (pH = 1) membrane showed high permeance (~275 GPU) and moderate selectivity (~23.5) at 150 °C and 1 bar. When the BTA solution was preadjusted to pH = 8, the gas permeance BIALP200 (pH = 8) membrane decreased and $H_2/CO_2$ selectivity increased (Supplementary Fig. 10a). This was because weak alkalinity of the BTA solution enhanced the reaction rate, resulting in higher crosslinking degree and denser membrane. In addition, the pressure-resistance ability of this membrane was weakened due to voids generated between the membrane layer and substrate (Supplementary Fig. 10b). BIALP200 (pH = 13) membrane

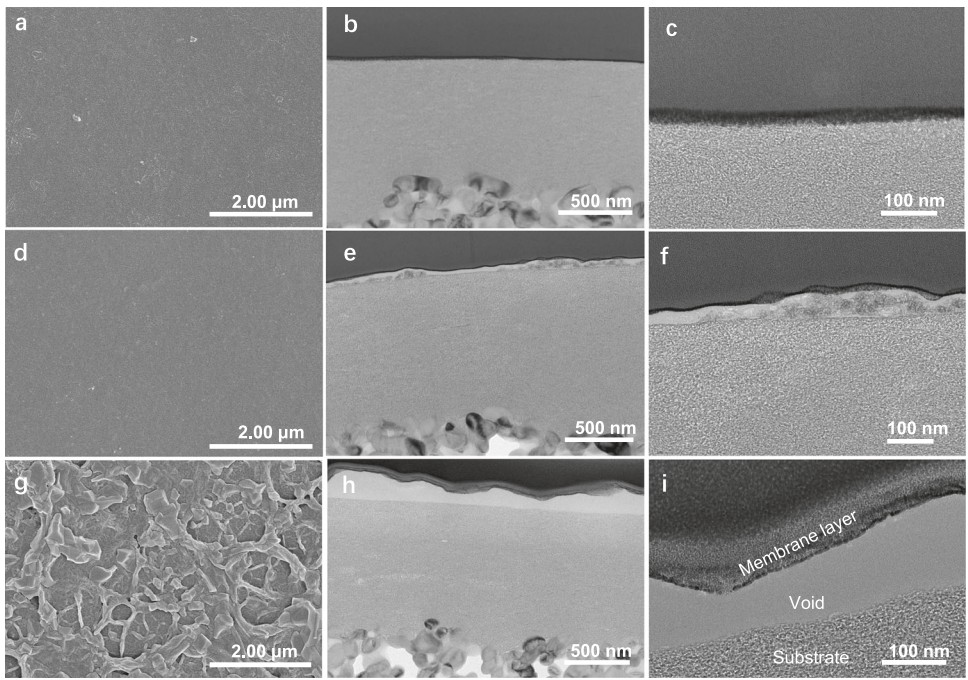

**Fig. 3 | SEM and TEM images of BIALP membranes prepared with pH variation of aqueous solutions. a** SEM image of top-surface BIALP200 (pH = 1) membrane. **b**, **c** TEM images of cross-section BIALP200 (pH = 1) membranes. **d** SEM image of top-surface BIALP200 (pH = 8) membrane. **e**, **f** TEM images of cross-section BIALP200 (pH = 8) membranes. **g** SEM image of top-surface BIALP200 (pH = 13) membrane. **h**, **i** TEM images of cross-section BIALP200 (pH = 13) membranes. The "membrane layer", "void", and "substrate" are marked in Fig. 3i. The same goes for other TEM images are not indicated to avoid redundancy.

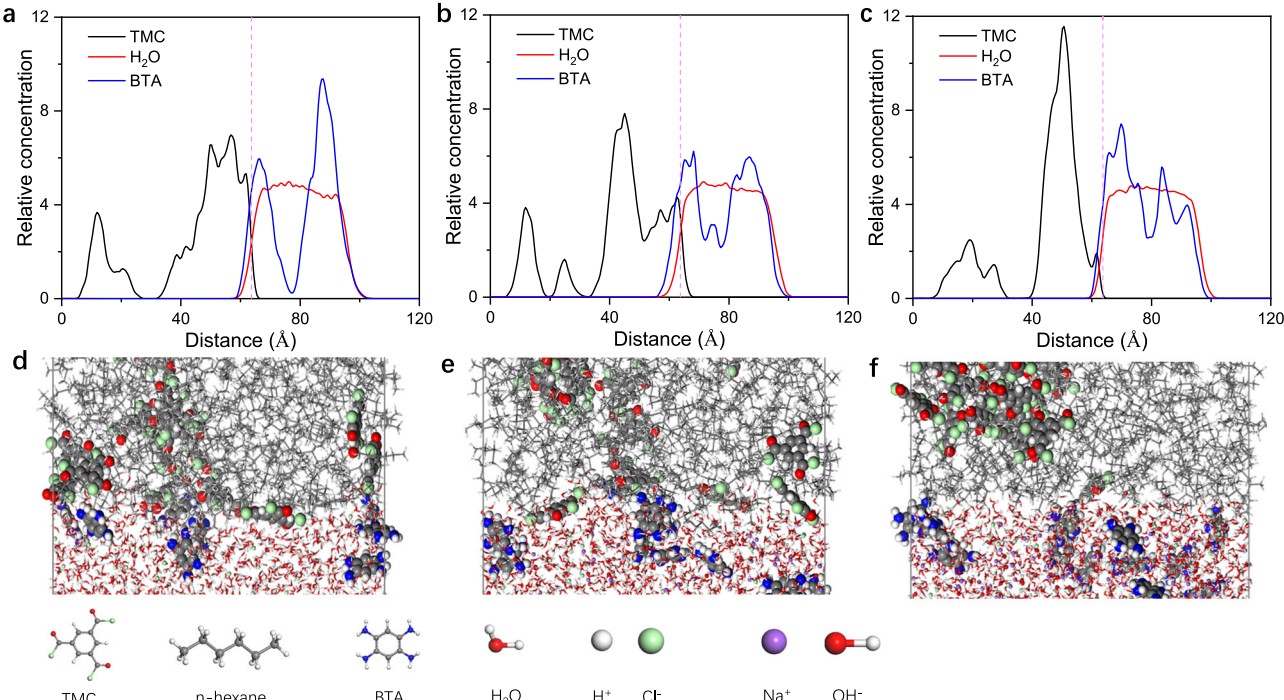

**Fig. 4 | MD simulation results of interfacial diffusion. a–c** Final (80 ps) relative abundance of BTA (blue curve), water (red curve), and TMC (black curve) near the interfaces (pink dashed line) with pH values of 1, 8, and 13 of aqueous solutions, respectively. **d–f** Final snapshots (80 ps) near the water/n-hexane interfaces with pH values of 1, 8, and 13 of aqueous solutions, respectively.

manifested ultra-high $H_2/CO_2$ selectivity of 120 and excellent $H_2$ permeance of 320 GPU. A further increase in crosslinking degree could contribute to the ultra-high selectivity. The crumpled surface provided a higher actual surface area and thus higher gas permeance.

## $H_2/CO_2$ separation performance

Nowadays, $H_2$ is generated mostly through fossil fuels by water-gas shift, which produces a mixture of $H_2$ (~55%) and $CO_2$ (~40%) at high temperatures (180–350 °C) and high-pressure (>10 bar)[39]. $H_2/CO_2$ separation using polymeric membrane-based molecular sieving is a big

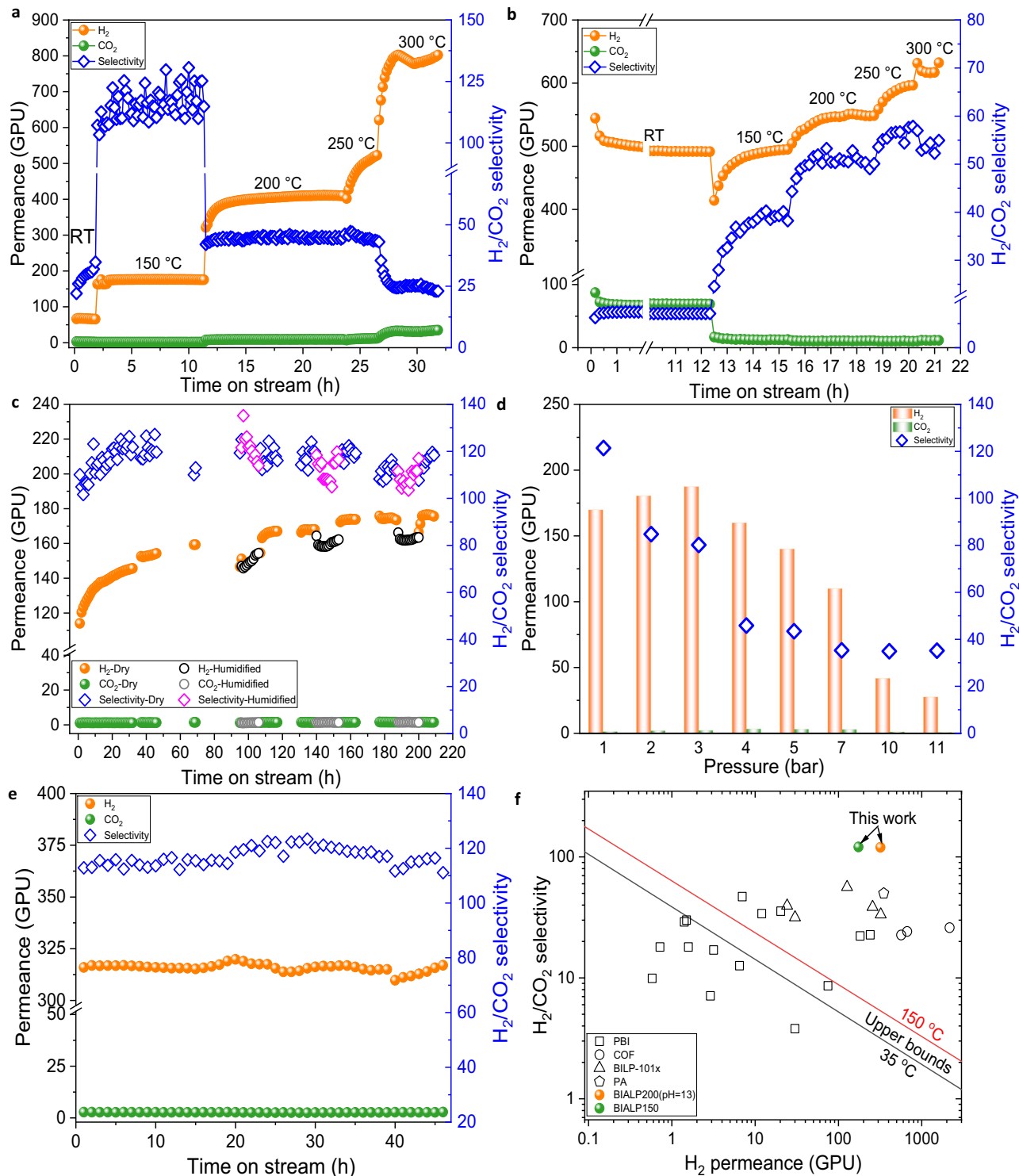

**Fig. 5 | H₂/CO₂ separation by ultra-thin BIALP membranes.** Effects of testing temperatures on the separation performance of **a** BIALP150 (pH = 1) membrane, **b** BIALP300 (pH = 1) membrane. Testing temperatures increased from RT (room temperature) to 300 °C. **c** Stability of BIALP150 (pH = 1) membrane for separating dry and humidified (2.3 mol % H₂O) feed. **d** Effect of feed pressure on the performance of BIALP150 (pH = 1) membrane. **e** Stability of BIALP200 (pH = 13) membrane. **f** Comparison of membrane performance. Upper bounds at 35 °C and 150 °C[7,29] are shown for comparison. Data in Supplementary Table 3 were used to make this plot. Feed conditions for Fig. 5: equimolar H₂ and CO₂, 150 °C, 1 bar (if not specified).

challenge owing to their difference of kinetic diameter as low as 0.4 Å[40]. Ultra-thin BIALP membranes were synthesized with improved sieving capability. The separation performances were evaluated with an equimolar, continuous flow H₂/CO₂ mixture as feed using the Wicke-Kallenbach method[20,22].

The effect of feed temperature was studied on BIALP150 (pH = 1), BIALP200 (pH = 1), and BIALP300 (pH = 1) membranes (Fig. 5a, b and Supplementary Fig. 11). During ~20 h of continuous tests, the feed temperature was raised gradually from room temperature (RT) to 300 °C. For BIALP150 (pH = 1) membrane (Fig. 5a), the H₂ permeance

reached 174 GPU at 150 °C with a remarkable selectivity of 121. Gas permeances of both $H_2$ and $CO_2$ increased with temperature indicating an activated diffusion for gas molecules. The increase of $CO_2$ permeance was greater than that of $H_2$, therefore, the selectivity fell. However, for the BIALP300 (pH = 1) membrane, at 150 °C, the permeance of $H_2$ dropped followed by a rise. Thermal movement of polymer chains, which could narrow the transport channels in this case, can be a possible reason. BIALP200 (pH = 1) and BIALP300 (pH = 1) membranes gave their highest selectivity of 33.4 and 57.9 at 250 °C, corresponding $H_2$ permeance of 374 and 592 GPU, respectively (Supplementary Fig. 11 and Fig. 5b). Higher temperature treatment made more ALP translate into BILP segments (Supplementary Fig. 2 and Supplementary Table 1). More benzimidazole linkages enhanced the conjugation effect between polymer chains, consequently, the packing of polymer chains was prearranged differently. That's why BIALP200 (pH = 1) and BIALP300 (pH = 1) membranes had better thermal resistance.

At 150 °C and 1 bar, the BIALP150 (pH = 1) membrane exhibited a superior separation performance with an $H_2/CO_2$ selectivity of 121 and an $H_2$ permeance of 176 GPU in a dry feed stream (Fig. 5c). When 2.3 mol% steam was introduced, both $H_2$ permeance and selectivity decreased slightly possibly due to the competitive permeation of water molecules in the membrane channels. After 210 h's measurements, the feed pressure was gradually increased to 11 bar (Fig. 5d). The fluctuation of permeance and selectivity could be explained by the interplay of the compaction effect of feed pressure and the swelling effect of $CO_2$. At 10 bar, the membrane still had good $H_2/CO_2$ selectivity (-35.8) and $H_2$ permeance (42 GPU), highlighting a good mechanical strength. Besides BIALP150 (pH = 1), a stability test of the BIALP200 (pH = 13) membrane was conducted (Fig. 5e), which manifested almost constant, exceptionally high $H_2$ permeance (around 315 GPU) and $H_2/CO_2$ selectivity (around 120). Supplementary Table 2 lists the synthetic parameters of BIALP membranes together with typical performances.

The performances of ultra-thin BIALP membranes surpassed not only the upper bound of conventional polymer membranes[41] but also that of a wide range of new polymer membranes (Fig. 5f, and Supplementary Table 3)[42–46]. For ALP membranes, the $H_2$ permeance was around 400 GPU, and $H_2/CO_2$ selectivity was only 1.9. The synergistic effect of sufficient intrinsic micropore and transient pores created by amide-linked segments and benzimidazole-linked parts (without overpacking of polymer chains) could well explain the high performance of BIALP membranes.

## Discussion

The kinetic diameters of $H_2$ and $CO_2$ are 0.29 and 0.33 nm, respectively. Their small size difference makes their separation challenging. In this work, intrinsic and transient sub-micropores were properly created in ultra-thin polymer membranes to achieve both high $H_2$ permeance and precise size discrimination of $H_2$ and $CO_2$. The key is the controllable regulation of the packing and connection of polymer chains during structural transformation. The transformation of ultra-thin membranes with high $H_2$ permeance and excellent $H_2/CO_2$ selectivity into practical applications has keen to research[21]. Therefore, the versatility of this approach is demonstrated via changing amine monomers (Supplementary Fig. 12) and substrates (Supplementary Fig. 13). This strategy will pave a new way for designing porous materials for separating light molecules with very small size differences.

## Methods
### Synthesis of powders
1,2,4,5-benzenetetramine tetrahydrochloride (BTA, Heowns, 98%) was dissolved in distilled water (Laboratory made) with a concentration of 1.5 wt%. A solution of 0.5 wt% trimesoyl chloride (TMC, TGI, > 98%) in n-hexane (Meryer, 99.5%) was poured onto the aqueous phase in a beaker. After 10 min of vigorous stirring, the powder was rinsed with N, N-Dimethylformamide (DMF, Meryer, >98%), and distilled water at least three times. After that, the ALP powders were put into convection ovens for heating under 150, 200, and 300 °C for 24 h forming BIALP150, BIALP200, and BIALP300 powders, respectively, and then naturally cooled down to room temperature.

### Preparation of freestanding films
Freestanding BIALP films were synthesized by interfacial polymerization and thermal treatment. BTA was dissolved in distilled water with a concentration of 1.5 wt%. A solution of 0.5 wt% TMC in n-hexane was slowly added to the aqueous solution and allowed to react for 10 min. Afterward, the resulting thin films were immediately picked up from the interface with a clean glass sheet, washed with ethanol, and transferred to a silicon wafer. The ALP thin films were put into convection ovens for heating under 150, 200, and 300 °C for 24 h, and BIALP150, BIALP200, and BIALP300 films were harvested.

### Fabrication of membranes
The BIALP membranes were fabricated by IP with BTA and TMC in aqueous and n-hexane phases, respectively, followed by thermal treatment. Asymmetric (γ-$Al_2O_3$ layer on α-$Al_2O_3$) porous ceramic disks 18 mm in diameter and 1 mm thick (Foshan Yirun Fine Ceramic New Material Co. Ltd.) were used as substrates. The average pore size of the top layers was around 5 nm. The ceramic disks were immersed in the 1.5 wt% of BTA aqueous solution for 20 min at room temperature. Excess droplets on top of the disks were removed using a piece of blotting paper from the back of the disks. Afterward, the disks were contacted with 0.5 wt% of TMC solution for 10 min under ambient conditions, and ALP membranes were formed on the surface of the disks. Then, the ALP membranes were immediately transferred to convection ovens for heating at 150, 200, and 300 °C for 24 h, to obtain BIALP150, BIALP200, and BIALP300 membranes, respectively.

### Fabrication of membranes with varying pH of aqueous solutions
The pH value of BTA aqueous solutions was preadjusted by NaOH. The ceramic disks were immersed in the 1.5 wt% BTA aqueous solutions for 30 min at room temperature. Excess droplets on top of the disks were removed using blotting papers from the back of the disks. Afterward, the disks were contacted with 0.5 wt% of a TMC solution for 10 min under ambient conditions. Then, the membranes were transferred to a convection oven and heated at 200 °C for 24 h. The membranes were labeled as BIALP200, BIALP200(pH = 8), and BIALP200 (pH = 13). Without the addition of NaOH, the pH value of aqueous solutions was 1. In this work, unless, otherwise stated, the pH value of the aqueous solution was 1.

### Characterizations
Fourier transform infrared spectra were measured in a range of 4000 to 500 cm$^{-1}$ using a Bruker tensor 2 infrared spectrometer. $^{13}$C ss-NMR spectra of $^{13}$C were performed on a JEOL JNM ECZ600R spectrometer. XPS spectra were obtained using an ESCALAB Xi+ Scientific spectrometer from ThermoFisher Scientific with Al-Kα as the X-ray source at a vacuum of $5 \times 10^{-8}$ Pa. The binding energy was calibrated with contaminated carbon C1s (284.8 eV). $N_2$ sorption isotherms were measured at – 196 °C and $CO_2$ sorption isotherms were measured at 25 °C by employing a BSD-PM (BeiShiDe Instrument) gas sorption analyzer. Prior to adsorption tests, the powdered samples were dried at 120 °C under a $N_2$ flow for 12 h. XRD was performed under ambient conditions with a Bruker-D8 Focus diffractometer operated at 40 mA and 40 kV using Cu Kα radiation with a step of 5 °/min from 5 to 50°. Environmental XRD was conducted at 150 °C and the specimens were surrounded by $H_2/CO_2$ (1:1, mol/mol) mixture. In-situ data were recorded at the 4th hour using Cu Kα radiation capture XRD signal with a step of 5 °/min from 5 to 50°. SEM was carried out using a field-emission gun

scanning electron microscope (HITACHI, S4800). Specimens were sputter-coated with a thin layer of gold before characterization. Thermal analyses were performed with a thermogravimetric analyzer (NETZSCH, TG 209 F3 Tarsus®). BIALP200 (pH = 1) and ALP powders were heated from room temperature to 600 °C at 5 °C/min in $N_2$. TMC and BTA powders were heated from room temperature to 600 °C at 5 °C/min in air. The density of BIALP200 (pH = 1) polymer solid was measured using a Micrometrics AccuPyc II1340 helium pycnometer equipped with a 3.5 cm³ chamber insert. The obtained values are the mean and standard deviation from a cycle of 5 measurements. Samples were evacuated thoroughly under vacuum at 120 °C for 24 h before measurements. AFM images were acquired from a Dimension icon (Brucker). The scans were performed in an air medium. The images were scanned in tapping mode using silicone cantilevers having a nominal diameter of less than 10 nm. Scanning was performed at a speed of 1.3 Hz, and a scan size of 5 μm was used for standard images. Bruker 'NanoScope Analysis beta' data visualization and analysis software were used to process the AFM images. To measure the thickness from AFM, freestanding films were transferred onto a silicon wafer. The thickness of the nanofilm was estimated from the height difference between the silicon and the upper surface of the nanofilm using a one-dimensional statistical function. Specimens (5 μm width, 3 μm height, ca. 70 nm thick) used in Fig. 3b, c, e, f, h, i were prepared by a focused ion beam (FEI Helios NanoLab 460HP) using 30 kV Ga ion, followed by further treatment to reduce the thickness. High-angle annular dark-field scanning transmission electron microscopy images were obtained using a FEI-Talos F200X STEM.

## Gas permeation test

The gas permeation test was performed on a homemade apparatus using the Wicke-Kallenbach technique. Each membrane was sealed in a membrane module with fluor rubber O-rings. The temperature of the module was controlled by a convection oven. Feed pressure was controlled by a pressure gauge. All the pressures presented in work are absolute pressures. The gas flow rate was regulated by mass flow controllers (MFCs). In general, the volume flow rate for each gas was set at 30 ml min⁻¹ on the feed side. Argon with atmospheric pressure was used as a sweep gas with a flow rate of 10 ml min⁻¹. Gas concentration in permeate was analyzed by a TCD detector on a gas chromatograph (Agilent GC 7890B).

Gas permeance ($P_i$) was calculated based on the following Eq. (1):

$$P_i = \frac{N_i}{A \times \triangle P_i} \qquad (1)$$

where $N_i$ refers to the permeate rate of component $i$ (mol s⁻¹), $A$ is the area of the effective membrane (m²), and $\triangle P_i$ is the pressure difference across the membrane of component $i$ (Pa).

Gas selectivity ($\alpha_{i/j}$) was defined as the ratio of their permeance by the following Eq. (2):

$$\alpha_{i/j} = \frac{P_i}{P_j} \qquad (2)$$

Where $P_i$ and $P_j$ are the gas permeance of permeate gas $i$ and $j$ (GPU). In this paper, gas permeance adopted a unit of gas permeation unit (GPU), and 1 GPU = 3.35 × 10⁻¹⁰ mol m⁻² s⁻¹ Pa⁻¹.

## Diffusion simulation

Amorphous cell modules in Materials Studio were used to simulate the trans-interface diffusion of BTA from water to hexane with the variation of pH and finial monomer concentration distributions were given. The chemical reaction between BTA and TMC molecules was not included in this simulated process. All systems were comprised of the same numbers of $H_2O$ (2200), BTA (22), and n-hexane (900) molecules

in a lattice cell (49 × 49 × 151 Å³). MD systems were simulated for 80 ps with NVE thermodynamic ensemble at 25 °C temperature.

## Data availability

The authors declare that the data supporting the findings of this study are available within the paper and its supplementary information file. All data are also available by request to the corresponding author. Source data are provided in this paper.

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

## Acknowledgements

The authors are grateful to the financial support from the National Key R&D Program of China (2021YFB3801200, Z.W.), the National Natural Science Foundation of China (22008171 and 21938007, X.L. and Z.W., respectively), the Inner Mongolia Autonomous Region Unveiling Project (2022JBGS0027, X.L.), the Seed Foundation of Tianjin University (2023XJD-0067, X.L.), the State Key Laboratory of Chemical Engineering (SKL-ChE-22B04, X.Y.) and the Haihe Laboratory of Sustainable Chemical Transformations.

## Author contributions

X.Y. and X.L. conceived the research idea and formulated the project. The experiment was performed by X.Y. including membrane fabrication and characterization, and performance evaluation. T.S. and X.Y. contributed to Fig. 1 conception and drawing. T.S. conducted MD simulations. M.L. carried out the SEM characterizations. Z.W. contributed to the gas sorption tests. The paper was written by X.Y. and revised by X.L. with input from all authors.

## Competing interests

The authors declare that they have no competing interests except for a Chinese patent (Application No. 2023101591417) filed through Tianjin University and invented by X.L. and X.Y.
