## [Peer Review File · Nature Communications]

Sub-micro porous thin polymer membranes for discriminating H₂ and CO₂Reviewers' Comments:

Reviewer #1:

Remarks to the Author:

The paper reports the synthesis of a novel porous polymer membrane through IP and its successful evaluation in H₂/CO₂ separation under practice-relevant conditions. Selectivity and hydrogen permeance are really superior. The referee is not a specialist in pore membranes, but the manuscript looks healthy and publication after minor revision is recommended.

Fig. 3 does not look exciting, rather boring and some information are missing. Maybe that the authors can make Fig. 3 more attractive and easier to understand for the readers of NatCom by using inscriptions. What is the 13-30 nm thick membrane layer, what is the support?

Apropos support, it can be accepted that the 13-30 nm thin polymer film prepared by IP is NOT self-supporting. Under "Fabrication of Membranes" on p.2 there is no information on "self-supporting" or "supported". Make this clearer in the manuscript at several positions.

In Fig. 2f, there are 3 curves. Explanation? In Fig. 2g, it would be helpful to explain to the reader what is Si, what is the polymer film. Use inscriptions.

Check, that every Figure is addressed in the text. I could not find Fig. 2d and Fig. 2h and others.

Reviewer #2:

Remarks to the Author:

Summary

In this work, the authors developed intrinsic and transient sub-micro pores polymer membranes with precise pore sizes by translating amine-linked polymer (ALP) films into benzimidazole-and-amine-linked polymer (BIALP) layers for H₂ and CO₂ separation. The fabrication protocol employed interfacial polymerization (IP) followed by thermal treatment, which appears to be a novel and straightforward approach. The features of the as-fabricated membrane are the thin (13-30 nm) selective layer and the controllable regulation of packing and connection of polymer chains during structural transformation. As a result, the membranes exhibited a high H₂ permeance and a high H₂/CO₂ selectivity, along with excellent stability under high pressures and temperatures. However, there are some concerns and issues that need to be satisfactorily addressed before it is suitable for publishing in Nature Communications.

List of detailed comments:

1. Page3, line 30, "provided a d-spacing of 3.3 Å at 150 °C and mixed H₂/CO₂ atmosphere (Fig. 2f)." It is confusing. What does "mixed H₂/CO₂ atmosphere" meaning?
2. Page5, lines 154-155, "The crumpled surface provided higher actual surface area and thus higher gas permeance." Gas transport/diffusion path is (or seems) perpendicular to the surface. The crumple surface may increase the overall surface area, but the effective diffusion area seems unchanged. How the crumple surface helps to boost the gas permeance? Where is the evidence of the crumple surface? Can you quantify the surface area or roughness?
3. Fig.3, please indicate/label which ones are SEM/TEM images, cross-section and top surface?
4. Fig. 1, -O and -NH₂ reaction/condensation, Why the other arm did not react?
5. Given a thickness of around 13 – 30 nm and a H₂ permeance of 315 GPU, the intrinsic permeability of the BIALP200 membrane is less than 10 Barrer which is among the lowest within polymeric materials while a high H₂/CO₂ selectivity slight over 100 is not too difficult to achieve using common crosslinking method and testing upon elevated temperatures. However, it is an advance in terms of fabrication technique or the thin film-forming property of the materials that allows for such an ultra-

thin film ($\ll 100$ nm) to form without defects and works stably. The authors are advised to put more emphasis on the ultra-thin film forming ability of their material/method in their writing.

6. Along the line of point 1, it would be important to know if such an ultra-thin film fabrication method/property is potentially transferable to other IP monomers such that this work would entail broader impact that caters to the readership of a general journal. For example, have the authors attempted other amine monomers other than BTA that could form a similar ultra-thin film like BIALP? How are the performance differed and what potentially cause that arising from the different chemical structure/property in the monomers?

7. In Fig 5, selectivity of BIALP150 decreased significantly from 150 °C onwards which is not observed on the BIALP300 membrane. Could the author explain this observation? And how is the BIALP200 membrane? Also decreasing selectivity above a certain temperature?

8. As a thin-film composite membrane that ultimately will require a scalable substrate to work, which is most likely to be polymeric in the industry and cannot withstand harsh thermal treatment, could the author comment on the scalability of their high thermal treatment temperature (150 – 300 °C) used in this work? Recent review sharing insights in this area could be added to the background section to support this part of discussion (J. Mater. Chem. A 11 (2023) 17452-17478)

9. Polymer based high H₂/CO₂ performance membrane material has recently been reported which be included in the literature review to enrich the background of this work (Small Methods 6 (2022) 2101288; Chemical Engineering Journal, (2023) 144073)

10. The ¹³C NMR spectra of the starting monomers should be provided for better comparison.

11. (On page 5, lines 148-155) It was explained that the gas permeance of the BIALP200 (pH=8) membrane decreased, and H₂/CO₂ selectivity increased because the weak alkalinity of the BTA solution enhanced the reaction rate, resulting in a higher crosslinking degree and a denser membrane. However, the BIALP200 (pH=13) membrane exhibited an ultra-high H₂/CO₂ selectivity and excellent H₂ permeance. Why does the gas permeance increase for the BIALP200 (pH=13) membrane as the crosslinking degree further increases? Why was the crumpled surface, which was expected to provide a higher actual surface area, not effective for enhancing gas permeance in the BIALP200 (pH=8) membrane?

12. (On page 3, lines 95-96) The authors stated that, due to the volatilization of more unreacted monomers and oligomers at higher temperatures, the thickness of BIALP membranes and films decreased from 30 to 13 nm. Please provide supporting evidence/references for this statement.

13. What is the duration of the interfacial polymerization (IP) process when forming an amine-linked polymer (ALP) film? What is the pore size of the BIALP membrane?

14. Can the authors provide the ratio of amide and benzimidazole linkages in the BIALP samples from NMR? Can the authors provide the density of BIALP membranes?

15. In Figure 5b, when the temperature increases from RT to 150 °C, why the permeance of H₂ decreases first and increases later?

16. Can the authors provide more evidence for the statement: "high temperature of thermal treatment could prearrange the packing of polymer chains."? What are the mechanical strength characteristics of these membranes.

Reviewer #3:

Remarks to the Author:

In the manuscript, high performance sub-micro porous polymer membranes were developed for H₂/CO₂ separation. The size difference of H₂ and CO₂ is only 0.04 nm. It is challenging to separate them based on size-selective diffusion, especially for polymer membranes. Surprisingly, the authors created membrane channels with precise molecular sieving capability. The key lies in the controllable regulation of packing and connection of polymer chains. Structural and process characterizations are sufficient. It is worthy of publication after a revision.

1. Recently, plenty of porous polymers were unveiled. To avoid confusion, the authors should clarify the relationship of porous organic frameworks (POFs), benzimidazole-linked polymers (BILPs), and the

newly-developed one benzimidazole-and-amine-linked polymer (BIALP).

2. BIALP membranes were transformed from ALP membranes. As evidenced, BIALP is H₂-selective. However, the selectivity of ALP membranes was only 1.9, even below Knudsen diffusion. What is the reason behind?

3. As presented in Fig. 2h, the membrane materials can adsorb CO₂ at 25 °C. Although the uptake will drop at elevated temperature, and high performance was delivered, the reviewer is curious. Are the membranes still performable at 25 °C?

4. Inorganic alumina disks were used as substrates. Did the authors try organic supports? If successful, it will be beneficial to the membrane economics.

5. Due to activated diffusion, gas permeance always increases with temperature. However, in fig. 5b, the permeance of H₂ at room temperature and 150 °C was comparable. Why? Explanations are wanted.

6. A number of membranes with different synthetic parameters, such as temperature of heat treatment, and pH of aqueous phase, were prepared. To make the work more readable, the reviewer encourages the authors to list the synthetic parameters in a table together with membrane names and typical performance.

7. Digital photos of membranes, films and powders can be present, to let readers catch the view of actual materials.

8. The membrane performance was compared with the upper bounds and polymeric membranes. Other membranes, for instance MOF, GO and MXene membranes, should be tabulated. Large advancement has been made in this area, very recently.

Black text: Comments from the editor and reviewers

Blue text: Responses

Red text: Revisions and highlights in the manuscript

Reviewer #1:

The paper reports the synthesis of a novel porous polymer membrane through IP and its successful evaluation in H₂/CO₂ separation under practice-relevant conditions. Selectivity and hydrogen permeance are really superior. The referee is not a specialist in pore membranes, but the manuscript looks healthy and publication after minor revision is recommended.

Response: Thanks for your positive evaluation of our work and the insightful comments.

1. Fig. 3 does not look exciting, rather boring and some information are missing. Maybe that the authors can make Fig. 3 more attractive and easier to understand for the readers of NatCom by using inscriptions. What is the 13-30 nm thick membrane layer, what is the support?

Response: As suggested, Fig.3 was reorganized. The “membrane layer”, “void”, and “substrate” were marked in Fig 3i. The same goes for other TEM images were not indicated to avoid redundancy. The cross-section SEM images were moved to Supplementary Fig. 7.

“Fig. 3 | SEM and TEM images of BIALP membranes prepared with pH variation of aqueous solutions. a, SEM image of top-surface BIALP200 (pH=1) membrane. b and c, TEM images of cross-section BIALP200 (pH=1) membranes. d, SEM image of top-surface BIALP200 (pH=8) membrane. e and f, TEM images of cross-section BIALP200 (pH=8) membranes. g, SEM image of top-surface BIALP200 (pH=13) membrane. h and i, TEM images of cross-section BIALP200 (pH=13) membranes. The “membrane layer”, “void”, and “substrate” are marked in Fig 3i. The same goes for other TEM images are not indicated to avoid redundancy.”

2. Apropos support, it can be accepted that the 13-30 nm thin polymer film prepared by IP is NOT self-supporting. Under “Fabrication of Membranes” on p.2 there is no information on “self-supporting” or “supported”. Make this clearer in the manuscript at several positions.

Response: The following description has been added in the section “Fabrication of membranes”. Moreover, we have checked the manuscript and made it clear.

“In details, ALP films and membranes were formed by reacting 1,2,4,5-benzenetetramine (BTA) with trimesoyl chloride (TMC) at the interface between aqueous and n-hexane phases (Fig. 1a and Supplementary Fig. 1). The films synthesized on the surface of substrates were named membranes, and photos of the films and BIALP membranes were shown in Supplementary Fig. 1.....In detail, preparation procedures were described in the Methods part.”

3. In Fig. 2f, there are 3 curves. Explanation? In Fig. 2g, it would be helpful to explain to the reader what is Si, what is the polymer film. Use inscriptions.

Response: The captions for Fig. 2f and 2g were rewritten. The “film” and “Si wafer” were marked in the Fig. 2g.

Fig. 2. ... f, Height profile and g, AFM image of BIALP200 film on silicon wafer. The three lines in f are corresponding by color to the lines in g....

4. Check, that every Figure is addressed in the text. I could not find Fig. 2d, Fig. 2h and others.

Response: Highly appreciated. All figures and tables have been noted and discussed in the revised manuscript.

Reviewer #2:

In this work, the authors developed intrinsic and transient sub-micro pores polymer membranes with precise pore sizes by translating amine-linked polymer (ALP) films into benzimidazole-and-amine-linked polymer (BIALP) layers for H₂ and CO₂ separation. The fabrication protocol employed interfacial polymerization (IP) followed by thermal treatment, which appears to be a novel and straightforward approach. The features of the as-fabricated membrane are the thin (13-30 nm) selective layer and the controllable regulation of packing and connection of polymer chains during structural transformation. As a result, the membranes exhibited a high H₂ permeance and a high H₂/CO₂ selectivity, along with excellent stability under high pressures and temperatures. However, there are some concerns and issues that need to be satisfactorily addressed before it is suitable for publishing in Nature Communications.

Response: We acknowledge the reviewer’s clear guidelines to improve the scientific significance of our manuscript.

List of detailed comments:

1. Page3, line 30, “provided a d-spacing of 3.3 Å at 150 °C and mixed H₂/CO₂ atmosphere (Fig. 2f).” It is confusing. What does “mixed H₂/CO₂ atmosphere” meaning?

Response: Environmental XRD characterization of BIALP200 (pH=1) powders was conducted at 150 °C and the specimens were surrounded by the H₂/CO₂ (1:1, mol/mol) mixture. The test conditions were described in the Methods part. The relative sentences in the manuscript have been updated.

“.....in a mixed H₂/CO₂ surrounding (Fig. 2e).....a 1 bar H₂/CO₂ (1/1, mol/mol) surrounding.....”

2. Page5, lines 154-155, “The crumpled surface provided higher actual surface area and thus higher gas permeance.” Gas transport/diffusion path is (or seems) perpendicular to the surface. The crumple surface may increase the overall surface area, but the effective diffusion area seems unchanged. How the crumple surface helps to boost the gas permeance? Where is the evidence of the crumple surface? Can you quantify the surface area or roughness?

Response: The membrane surface SEM images and cross-sectional TEM images confirmed that the membrane surface changed from smooth to crumpled (Fig. 3) with pH values of amine solutions. As Fig. R1a shown, for the membrane with a smooth surface, the gas transport pathway is perpendicular to the surface. However, for a crumpled membrane (Fig. R1b), effective gas transport pathways are increased with the area of selective membrane layer, resulting in higher gas permeance.

Fig. R1. Smooth and crumpled morphologies of membrane surfaces and gas transport pathways

(directions indicated by arrows). **a**, smooth surface. **b**, crumpled surface.

The BET area of BIALP200 (pH=1) powders was $21.5 \text{ m}^2 \text{ g}^{-1}$ based on N_2 adsorption (Fig. 2i), comprising just external surface area according to the t-plot method. As shown in the Supplementary Fig. 8, the roughness of membrane surface increased with pH values of amine solutions, in line with surface SEM images and cross-sectional TEM images of Fig. 3. In the main text, the description of surface roughness has been added to support this point of view.

Supplementary Figure 8. Three-dimensional morphologies and roughness of BIALP membranes. a, BIALP200 (pH=1) membrane. **b**, BIALP200 (pH=8) membrane. **c**, BIALP200 (pH=13) membrane.

“.....The roughness of membrane surface was increased from 5.30 to 9.65 nm (Supplementary Fig. 8a and b)..... relative high roughness of 20.7 nm (Supplementary Fig. 8c), and.....”

3. Fig.3, please indicate/label which ones are SEM/TEM images, cross-section and top

surface?

Response: Done. See our response to comment 1 from reviewer 1.

4. Fig. 1, -O and -NH₂ reaction/condensation, Why the other arm did not react?

Response: After thermal treatment, the ¹³C ss NMR spectrum of BIALP200 (pH=1) (Fig. 2c) showed that both amide (175 ppm, number 3) and benzimidazole (150 ppm, number 7) linkages existed. So, thermal treatment only partly transformed amide into benzimidazole linkages, verifying the claim of BIALP structure in Fig. 1a. This conclusion was also supported by XPS (Supplementary Table 1), where the atomic ratio of -N- higher than 50% was listed.

5. Given a thickness of around 13 – 30 nm and a H₂ permeance of 315 GPU, the intrinsic permeability of the BIALP200 membrane is less than 10 Barrer which is among the lowest within polymeric materials while a high H₂/CO₂ selectivity slight over 100 is not too difficult to achieve using common crosslinking method and testing upon elevated temperatures. However, it is an advance in terms of fabrication technique or the thin film-forming property of the materials that allows for such an ultra-thin film (<< 100 nm) to form without defects and works stably. The authors are advised to put more emphasis on the ultra-thin film forming ability of their material/method in their writing.

Response: We acknowledge the reviewer's advice to improve the manuscript. The advantage of our BIALP membranes on ultra-thin thickness has gained more emphasis in the main text.

“Herein, sub-micro porous ultra-thin (down to 13 nm) polymer membranes were fabricated...”, “Ultra-thin BIALP membranes were synthesized with improved sieving capability.”

“The BIALP membranes showed ultra-thin thickness thanks to the combined action of the fast reaction rate of monomers, small pore size of substrates, and post treatment at high temperatures.”

6. Along the line of point 1, it would be important to know if such an ultra-thin film

fabrication method/property is potentially transferable to other IP monomers such that this work would entail broader impact that caters to the readership of a general journal. For example, have the authors attempted other amine monomers other than BTA that could form a similar ultra-thin film like BIALP? How are the performance differed and what potentially cause that arising from the different chemical structure/property in the monomers?

Response: Done. See below the newly-added Supplementary Fig. 12 and descriptions.

Supplementary Figure 12. The versatility of the membrane synthetic approach via using the other amine monomer. a, Reaction scheme for forming a DAB-BIALP membrane. The three segments on the right part can be possibly connected to the numbered positions in the middle part. **b**, SEM image (inset, digital photo) of top-surface DAB-BIALP150 (pH=1) membrane. **c**, SEM image of cross-section DAB-BIALP150 (pH=1) membrane. **d**, Stability of a DAB-BIALP150

(pH=1) membrane. Feed conditions: equimolar H₂ and CO₂, 150 °C, 1 bar. **d**, Effect of feed pressure on the performance of a DAB-BIALP150 (pH=1) membrane. Feed conditions: equimolar H₂ and CO₂, 150 °C.

To confirm the versatility of this fabrication method, 3,3'-diaminobenzidine (DAB) monomer was used instead of BTA using the same procedure as BIALP150 (pH=1) membranes. The membranes were noted as DAB-BIALP150 (pH=1) membranes. As shown in Supplementary Figure 12c, the membrane had a similar ultra-thin thickness, roughly 40 nm.

The H₂/CO₂ separation performance of a DAB-BIALP150 (pH=1) membrane was tested at 150 °C, 1 bar (Supplementary Fig. 12d, e). Results confirmed that the H₂/CO₂ selectivity was around 70 corresponding to a H₂ permeance of 108 GPU. Furthermore, this membrane exhibited good pressure-resistance. Compared with the BIALP150 (pH=1) membranes, both the H₂ permeance and H₂/CO₂ selectivity were slightly lower. This is probably because the use of longer DAB monomer generated longer tightly-packed polymer chains, reducing the transport channels of H₂.

7. In Fig 5, selectivity of BIALP150 decreased significantly from 150 °C onwards which is not observed on the BIALP300 membrane. Could the author explain this observation? And how is the BIALP200 membrane? Also decreasing selectivity above a certain temperature?

Response: The effect of feed temperature on the performance of a BIALP200 (pH=1) membrane had been shown in Supplementary Fig. 11. Both the observation, and the explanation of these three membranes had been elaborated in the second paragraph of “H₂/CO₂ separation performance” section.

8. As a thin-film composite membrane that ultimately will require a scalable substrate to work, which is most likely to be polymeric in the industry and cannot withstand harsh thermal treatment, could the author comment on the scalability of their high thermal treatment temperature (150 – 300 °C) used in this work? Recent review sharing insights in this area could be added to the background section to support this part of discussion (J. Mater. Chem. A 11 (2023) 17452-17478)

Response: Considering the requirement of large-scale commercial developments, both inorganic and organic substrates were used in this work. BIALP membranes could be formed on polymeric substrates, such as polyacrylonitrile (PAN), after a thermal treatment at 150 °C. The membrane showed a H₂/CO₂ selectivity of 62 corresponding to a H₂ permeance of 738 GPU at 150 °C (Supplementary Fig. 13).

The following statements have been supplemented in the background and outlook sections. The comprehensive review paper has been cited.

“..., the membrane performance and preparation technology should be further improved to meet industrial requirements²¹.”

“The transformation of ultra-thin membranes with high H₂ permeance and excellent H₂/CO₂ selectivity into practical applications is keen to be investigated²¹. Therefore, the versatility of this approach is demonstrated via changing amine monomers (Supplementary Fig. 12) and substrates (Supplementary Fig. 13).”

9. Polymer based high H₂/CO₂ performance membrane material has recently been reported which be included in the literature review to enrich the background of this work (Small Methods 6 (2022) 2101288; Chemical Engineering Journal, (2023) 144073)

Response: These two research articles introduced a strategy to design gas-selective channels. Organic macrocyclic cavitands were incorporated into scalable polymers to create open nanocavities with strong size-sieving for enhancing H₂/CO₂ selectivity. This relative strategy has been discussed and cited to enrich the background of our work.

“.....using approaches such as stacking 2D porous layers¹⁰, integrating aligned synthesis^{20,21}, introducing nanocavities^{18,19}, and post-treatment²⁴ to design channels.”

10. The ¹³C NMR spectra of the starting monomers should be provided for better comparison.

Response: Done. Compared with the starting monomers (Supplementary Fig. 3), the formation of amide and benzimidazole linkages was corroborated by the new feature

peaks in ^{13}C ss-NMR spectra (Fig. 2c) at 175 ppm and 150 ppm, respectively.

Supplementary Figure 3. ^{13}C NMR spectra of BTA and TMC monomers. **a**, ^{13}C solid-state NMR of the BTA. **b**, ^{13}C liquid-state NMR of the TMC (DMSO- d_6 was used as solvent).

11. (On page 5, lines 148-155) It was explained that the gas permeance of the BIALP200 (pH=8) membrane decreased, and H_2/CO_2 selectivity increased because the weak alkalinity of the BTA solution enhanced the reaction rate, resulting in a higher crosslinking degree and a denser membrane. However, the BIALP200 (pH=13) membrane exhibited an ultra-high H_2/CO_2 selectivity and excellent H_2 permeance. Why does the gas permeance increase for the BIALP200 (pH=13) membrane as the crosslinking degree further increases? Why was the crumpled surface, which was expected to provide a higher actual surface area, not effective for enhancing gas permeance in the BIALP200 (pH=8) membrane?

Response: The BIALP200 (pH=13) membranes had crumpled surface (Fig. 3g), and thus higher actual surface area, so their permeance was augmented. However, the surface of BIALP200 (pH=8) membranes was smooth (Fig. 3d), and a lower permeance was understandable.

12. (On page 3, lines 95-96) The authors stated that, due to the volatilization of more unreacted monomers and oligomers at higher temperatures, the thickness of BIALP membranes and films decreased from 30 to 13 nm. Please provide supporting evidence/references for this statement.

Response: As mentioned in the section “Methods”, the ALP membranes formed were

immediately transferred to preheated convection ovens for thermal treatment. The following TGA results demonstrated that monomers used in this work exhibited significant weight loss before 300 °C (Supplementary Fig. 6). Specifically, TMC completely volatilized at around 250 °C. In addition, there was an obvious weight loss before 300 °C for ALP powder due to structural transformation and oligomer volatilization. These results provide sufficient evidence for the volatilization of unreacted monomers and oligomers. So, the further growth of membrane was greatly inhibited at higher temperatures, for instance, 300 °C, due to the lack of nutrition, interpreting the change of membrane thickness from 30 to 13 nm.

Supplementary Figure 6. TGA curves of monomers, ALP and BIALP powders. BIALP200 (pH=1) (orange line) and ALP (green line) powders were heated from room temperature to 600 °C at 5 °C/min in N₂. TMC (red line) and BTA (black line) monomers were heated from room temperature to 600 °C at 5 °C/min in air.

13. What is the duration of the interfacial polymerization (IP) process when forming an amine-linked polymer (ALP) film? What is the pore size of the BIALP membrane?

Response: As the section “Methods” mentioned, the aqueous solution was contacted with organic solution for 10 min under ambient conditions, and ALP membranes were formed.

The pore size of BIALP200 powders was below 0.36 nm since N₂ molecules (kinetic diameter 0.36 nm) had no access to the narrow micropores in these polymers (Fig. 2i). CO₂ (kinetic diameter 0.33 nm) adsorption showed the microporous surface area of BIALP200 was 58.9 m² g⁻¹, highlighting the existence of sub-micro pores in its internal structure. The discussion had been provided in the fourth paragraph of section

“Structural properties of membranes”.

14. Can the authors provide the ratio of amide and benzimidazole linkages in the BIALP samples from NMR? Can the authors provide the density of BIALP membranes?

Response: The ^{13}C ss-NMR used here can provide qualitative analysis instead of quantitative data.

“Density of BIALP200 (pH=1) polymer solid was measured using a Micrometrics AccuPyc II 1340 helium pycnometer equipped with a 3.5 cm³ chamber insert. The obtained values are the mean and standard deviation from a cycle of 5 measurements. Samples were evacuated thoroughly under vacuum at 120 °C for 24 h before measurements.”

The information has been involved in the fourth paragraph of section “Structural properties of membranes”. “with a density of $1.56 \pm 0.005 \text{ g cm}^{-3}$,.....”

15. In Figure 5b, when the temperature increases from RT to 150 °C, why the permeance of H₂ decreases first and increases later?

Response: The reason has been added.

“However, for the BIALP300 (pH=1) membrane, at 150 °C, the permeance of H₂ dropped followed by a rise. Thermal movement of polymer chains, which could narrow the transport channels in this case, can be a possible reason.”

16. Can the authors provide more evidence for the statement: “high temperature of thermal treatment could prearrange the packing of polymer chains.”? What are the mechanical strength characteristics of these membranes.

Response: The chemical structural transformation from ALP to BIALP after thermal treatment was evidenced by XPS and NMR (Fig. 2 a-c). The prearrangement of the packing of polymer chains was verified by the enhanced sub-micro pores (Fig. 2h). “Higher temperature treatment made more ALP translate into BILP segments (Supplementary Fig. 2 and Supplementary Table 1). More benzimidazole linkages enhanced the conjugation effect between polymer chains, consequently, the packing of polymer chains was prearranged differently. That’s why BIALP200 (pH=1) and

BIALP300 (pH=1) membranes had better thermal resistance.”

“.....Both ALP and BIALP films could be handled, but teared easily. A video of an ALP film pierced by a pipette was attached in the supplementary materials.....”

“.....At 10 bar, the membrane still had good H₂/CO₂ selectivity (~35.8) and H₂ permeance (42 GPU), highlighting a good mechanical strength.....”

Reviewer #3:

In the manuscript, high performance sub-micro porous polymer membranes were developed for H₂/CO₂ separation. The size difference of H₂ and CO₂ is only 0.04 nm. It is challenging to separate them based on size-selective diffusion, especially for polymer membranes. Surprisingly, the authors created membrane channels with precise molecular sieving capability. The key lies in the controllable regulation of packing and connection of polymer chains. Structural and process characterizations are sufficient. It is worthy of publication after a revision.

Response: The reviewer’s strong recommendation and invaluable suggestions are highly appreciated.

1. Recently, plenty of porous polymers were unveiled. To avoid confusion, the authors should clarify the relationship of porous organic frameworks (POFs), benzimidazole-linked polymers (BILPs), and the newly-developed one benzimidazole-and-amine-linked polymer (BIALP).

Response: The statement below has been added to the introduction part.

“POFs are porous, organic, network polymers linked by covalent bonds²². Benzimidazole-linked polymers (BILPs) are a family of POFs, the linkage of which is benzimidazole²²..... BIALPs are distinct from BILPs as amine linkages are present in the former ones.”

References:

22. Shan, M., Liu, X., Wang, X., Yarulina, I., Seoane, B., Kapteijn, F., Gascon, J., Facile manufacture of porous organic framework membranes for precombustion CO₂ capture. *Sci. Adv.*, **4**, eaau1698 (2018).

2. BIALP membranes were transformed from ALP membranes. As evidenced, BIALP is H₂-selective. However, the selectivity of ALP membranes was only 1.9, even below Knudsen diffusion. What is the reason behind?

Response: ALP membranes were fabricated by IP with BTA and TMC in aqueous and n-hexane phases, respectively. Pinholes might be still remaining after the room temperature IP. Therefore, these ALP membranes could not select H₂ from CO₂. In the following immediate thermal treatment, unreacted monomers around the pinholes could quickly diffuse and cover the bare areas of substrates, then, reacted to form membranes to fill these pinholes. So, BIALP membranes were selective.

3. As presented in Fig. 2h, the membrane materials can adsorb CO₂ at 25 °C. Although the uptake will drop at elevated temperature, and high performance was delivered, the reviewer is curious. Are the membranes still performable at 25 °C?

Response: The performances of BIALP membranes were listed in Supplementary Table 2. Results showed that these membranes could separate H₂ and CO₂ at 25 °C (R.T.) and 1 bar.

Supplementary Table 2. Synthetic parameters and separation performances of membranes studied in this work.

Membrane	Synthetic parameters		Test parameters			
	pH	Heating temperature (°C)	Temperature (°C)	Pressure (bar)	P_{H_2} (GPU)	α_{H_2/CO_2}
BIALP150	1	150	R.T.	1	65.4	29.1
			150	1	176	121
			200	1	405	44.3
			250	1	517	43.6
			300	1	795	25.1
			150	2	180	84.8
			150	3	187	80.2
			150	4	160	45.9

			150	5	140	43.4
			150	7	110	35.5
			150	10	41.9	35.8
			150	11	27.8	35.6
			R.T.	1	145	9.63
			150	1	302	24.9
BIALP200	1	200	200	1	321	25.3
			250	1	402	30.2
			300	1	750	33.6
			R.T.	1	502	7.01
			150	1	504	40.2
BIALP300	1	300	200	1	550	53.5
			250	1	597	58.1
			300	1	610	69.4
			R.T.	1	46.5	59.9
			150	1	63	121
			150	3	45	147
			150	5	42	105
			150	7	44	61.6
BIALP200	8	200	150	10	27	23.4
			150	11	51	12.0
			150	12	58	8.3
			150	13	120	6.7
			150	14	135	7.3
			R.T.	1	302	32.1
BIALP200	13	200	150	1	315	120

Feed stream: H₂/CO₂ (1/1, mol/mol) mixed gas.

R.T.: Room temperature.

4. Inorganic alumina disks were used as substrates. Did the authors try organic supports?

If successful, it will be beneficial to the membrane economics.

Response: Done.

Supplementary Figure 13. BIALP150 (pH=1) membranes formed on PAN substrates. a, SEM image (inset, digital photo) of surficial BIALP150 (pH=1) membrane on PAN substrate. **b**, SEM image of cross-section BIALP150 (pH=1) membrane on PAN substrate.

Commercial polyacrylonitrile (PAN) membranes (Beijing Separate Equipment Co., Ltd, 50 KDa) were also selected as substrates. BIALP150 (pH=1) membranes were formed via the IP process mentioned in the main text followed by a thermal treatment at 150 °C. The SEM images and a digital photo of a BIALP membrane on PAN substrate are shown in Supplementary Fig. 13. The membrane showed a H₂/CO₂ selectivity of 62.0 corresponding to a H₂ permeance of 738 GPU at 150 °C and 1 bar.

5. Due to activated diffusion, gas permeance always increases with temperature. However, in fig. 5b, the permeance of H₂ at room temperature and 150 °C was comparable. Why? Explanations are wanted.

Response: See our response to comment 15 from Reviewer 2.

6. A number of membranes with different synthetic parameters, such as temperature of heat treatment, and pH of aqueous phase, were prepared. To make the work more readable, the reviewer encourages the authors to list the synthetic parameters in a table together with membrane names and typical performance.

Response: Thanks. Synthetic parameters and separation performances of membranes studied in this work were listed in Supplementary Table 2.

7. Digital photos of membranes, films and powders can be present, to let readers catch the view of actual materials.

Response: Digital photos of membranes (Supplementary Fig. 1e and Fig. 13), films, and powders (Supplementary Fig. 1c, d) were provided in the supplementary information.

8. The membrane performance was compared with the upper bounds and polymeric membranes. Other membranes, for instance MOF, GO and MXene membranes, should be tabulated. Large advancement has been made in this area, very recently.

Response: As suggested, we summarized different membranes for H₂/CO₂ separation in the supplementary information.

Supplementary Table 4. Recently-reported other membrane performances for H₂/CO₂ separation. The feed is H₂ and CO₂ mixture.

Membrane	Operation conditions		Thickness (μm)	PH_2 (GPU)	α_{H_2/CO_2}	Ref.	
	Temperature ($^{\circ}\text{C}$)	Pressure (bar)					
Co-gallate MOF	150	1.1	0.556	150	60	17	
[Cu ₂ Br(IN) ₂] _n	25	-	<0.01	527	294	18	
Zn ₂ (bim) ₄	25	1	0.001	2700	291	19	
NH ₂ -Zn ₂ (bim) ₄	R.T.	1	0.25	1095	1542	20	
Zn ₂ (bim) ₄	R.T.	1	1*	1417	1158	21	
Zn ₂ (bim) ₄	20	1	<0.01	2320	166	22	
(Zn/Co) ₂ (bim) ₄	150	1	0.08	630	70	23	
ZIF-8	25	1	970	2654	17	24	
ZIF-95	25	2	0.4-0.5	571	184	25	
ZIF-95	200	1	20	493	41.6	26	
ZIF-95	R.T.	1	0.6	2291	32.2	27	
b-ZIF-L	240	1	25	1330	989	28	
MOF ZIF-L	25	1	0.04	4033	321	29	
bMOF201	R.T.	1	62.7	1877	22.2	30	
NH ₂ -UiO-66	25	1	0.18	1230	41.3	31	
NH ₂ -UiO-66	20	1	4	1039	28.2	32	
UiO-67	R.T.	1	0.2	1809	8.9	33	
NH ₂ -MIL-53(Al)	80	1	15	686	23.7	34	
NH ₂ -MIL-125(Ti)	30	1	0.5	129.3	22.8	35	
MAMS-1	40	1	0.04	880	225	36	
IRMOF-3	25	-	0.004	971	1473	37	
Ni-LAP	120	1	4.3	298	35	38	
KAUST-7	25	1	1-2	638	17.7	39	
KAUST-7	25	1	20	1537	27.3	40	
GO	GO	25	-	0.05	10106	4	41

	q-rGO	25	-	1	783	3636	42
	GO	120	-	0.002	2000	47	43
	GO	20	-	0.009	344	3400	44
	MXene	25	1	2	1113	167	45
	MXene	25	1	0.02	1200	20	46
MXene	MXene	R.T.	1	0.22	70.6	30.3	47
	Pd-MXene	25	-	0.78	794	242	48
	MXene-ZIF-8	25	-	0.45	178	77	49
	RUB-15	250	1	0.3	848	154	50
Zeolites	RUB-15	225	2	0.15	160	30	51
	NaA	25	2	12.4	406	7.1	52

Reviewers' Comments:

Reviewer #1:

Remarks to the Author:

the authors did a proper revision. From my side: accept.

PS As indication that I carefully studied the revised manuscript: On p. 5 a space is missing in "Fig.3a".

Reviewer #2:

Remarks to the Author:

The authors have carefully answered and addressed the questions raised by the reviewers. The manuscripts have been revised and improved accordingly. It is suitable for publication. However, there is a minor suggestion for the authors: in the Supplementary Table 4, referring to entry ZIF-8 (ref. 24), the TFC thickness is 970 μm and has H_2 permeance 2654 GPU and $\text{H}_2/\text{CO}_2 = 17$. It is unlikely that such thick membrane has so high permeance and selectivity. Please check and verify it.

Reviewer #3:

Remarks to the Author:

The authors have addressed all my concerns. It can be published in Nature Communications.

Black text: Comments from the editor and reviewers

Blue text: Responses

Red text: Revisions and highlights in the manuscript

Reviewer #1:

the authors did a proper revision. From my side: accept.

PS As indication that I carefully studied the revised manuscript: On p. 5 a space is missing in "Fig.3a".

Response: Done.

Reviewer #2:

The authors have carefully answered and addressed the questions raised by the reviewers. The manuscripts have been revised and improved accordingly. It is suitable for publication. However, there is a minor suggestion for the authors: in the Supplementary Table 4, referring to entry ZIF-8 (ref. 24), the TFC thickness is 970 μm and has H_2 permeance 2654 GPU and $\text{H}_2/\text{CO}_2 = 17$. It is unlikely that such thick membrane has so high permeance and selectivity. Please check and verify it.

Response: We checked and verified that the thickness of the ZIF-8 membrane was 970 μm . It is a free-standing membrane.

Reviewer #3:

The authors have addressed all my concerns. It can be published in Nature Communications.

Response: Thanks for your recommendation.